# Spectroscopic measurement of near-infrared soil pH parameters based on GhostNet-CBAM

**Jianguo Zhu[1,2], Wenjin Wang[2,3], Peng Tian** [ID][2,3]*

**1** School of Information Science and Engineering, Hunan Institute of Science and Technology, Yueyang, Hunan Province, China, **2** Key Laboratory of Hunan Province on Information Photonics and Freespace Optical Communications, Yueyang, Hunan Province, China, **3** School of Physics and Electronics, Hunan Institute of Science and Technology, Yueyang, Hunan Province, China

* tianpp815@163.com

**Data availability statement:** The data can be found in https://esdac.jrc.ec.europa.eu/resource-type/european-soil-database-soil-properties.

## Abstract

Soil pH is an important parameter that affects plant nutrient uptake and biological activity and has received extensive attention and research. In this paper, we propose a neural network algorithm using Ghostnet combined with Convolutional Block Attention Module (CABM) to realize the near-infrared (NIR) PH spectral measurement of soil. The method firstly utilizes Monte Carlo Cross Validation (MCCV) method to reject the anomalous samples in the data, and then uses GhostNet combined with CBAM algorithm to train and predict the PH values of the four Lucas soil spectral data measured by the two different methods, and compares the prediction results with those of PLSR and VGGNet-16 methods. The results showed that the $R^2$ of the GhostNet-CBAM method could reach up to 0.9447, and the RMSE reached as low as 0.3267, and the scatter density plots of the predicted and true values further confirmed that the method could quickly and accurately obtain the soil pH parameters.

## Introduction

Soil fertility is influenced by acidity and alkalinity and is an important indicator of the ability of a soil to provide sufficient essential nutrients for normal crop growth and development [1]. Spectral analysis of visible-near-infrared (VNIR, 400–2500 nm) reflectance and transmittance properties of soils has enabled effective extraction and measurement of soil pH characteristics [2]. VNIR spectra are the result of radiation absorption by various chemical bonds (e.g., C-H, N-H, and O-H) in soil samples [3,4]. The pH parameter can be quantitatively determined from the correlation of the VNIR spectra with the main spectrally active components.

To extract quantitative information from VNIR spectra, multivariate regression models can be used. David A. Laird used Principal Component Regression (PCR) to predict soil properties [5]. Wenjun Ji used the CSSL model to predict pH in rice soil [2]. Xian-Zhong Shi et al. used the Partial Least Squares Regression (PLSR) algorithm to model soil hyperspectral data and successfully applied it to HyMap images to derive an acidity map [6].

**Funding:** This research was funded by the Key Research Foundation of Hunan Province Education Department (No. 21A0404).

In addition, some researchers have applied machine learning methods in modeling spectral analysis, Songchao Chen et al. [7] used Cubist model, S. Nawar et al. [8] and Shancai Xiao et al. [9] used Random Forest (RF) model, Chiranjit Singha et al. [10] used the Support Vector Machine Regression (SVMR) model, Florian Huber et al. [11] and Shengxiang Xu et al. [12] used the Extreme Gradient Boosting (XGBoost) model for soil VNIR spectral analysis.) model to model soil VNIR spectra to predict soil properties. However, since VNIR data contains thousands of wavelengths with strong covariance and complex relationships among them, traditional machine learning methods have limited capability in processing these data. These methods require extensive pre-processing of the spectra to extract characteristic wavelengths associated with soil properties.

With the advancement of computer technology, many scholars have proposed solutions to extract quantitative information from VNIR spectra using deep learning methods. Veres et al. [13] applied one-dimensional convolutional neural network (1D-CNN) techniques to soil spectroscopy for the first time, demonstrating that deep learning is particularly effective for estimating large soil properties. Liu et al. [6] applied another 1D-CNN with a different architecture to predict soil properties in the Lucas database. Minasny et al. [14] converted the initial spectra into 2D spectrograms in order to apply a 2D Convolutional Neural Network (2D-CNN). Transfer learning has also been used to localize global models in different studies. Finally, Riese and Keller [15] applied another 1D-CNN on the same dataset in order to classify the texture of each soil sample using the German soil texture classes. Tsakiridis et al. [16] developed a localized multichannel 1D-CNN for the prediction of 10 soil physical and chemical properties from the LUCAS spectral library. In addition, Xi Guo et al. [17] used a deep convolutional neural network (DCNN) to model large-scale soil spectral data to predict multiple soil parameters. Wang et al. [18] proposed a model based on multi-gate hybrid expert network (MMoE), which can improve the prediction accuracy of soil parameters by combining with data enhancement. Compared with the traditional single- and multi-task models, MMoE reduces the RMSE by 5%–48% and improves the R2 by 1%–119%. Li et al. [19] proposed a lightweight deep learning model, DSCformer, to predict soil parameters, which combines the Metaformer architecture with deep separable convolution to reduce the computational complexity and improve the accuracy, and analyzes the interpretability with the help of SHAP method, which specifies the spectral bands related to soil nitrogen content. It analyzes the interpretability with the help of SHAP method and clarifies the spectral bands related to soil nitrogen content.

Due to the poor predictive accuracy of machine learning in predicting large soil spectral data, the large number of model parameters of most deep learning algorithms is not conducive to on-the-ground applications. In order to simultaneously ensure a lower number of model parameters and higher prediction accuracy, a novel GhostNet-CBAM combined model is proposed in this paper. GhostNet is a lightweight model [20] that is primarily suited for image classification tasks [21], but is modified appropriately to allow it to predict soil parameters. The model greatly reduces the number of model parameters while maintaining accuracy by combining ordinary convolution and simple linear operations. Convolutional Block Attention Module (CBAM) [22] enhances the feature extraction capability of the model by introducing only a small number of additional parameters through channel attention and spatial attention mechanisms. In this paper, the GhostNet-CBAM model was used to model the raw spectral data and the data processed by the MCCV algorithm to predict the pH parameters of the soil, and the effect of CBAM on the enhancement of model features was analyzed. Then it is compared with the traditional regression algorithm and the VGGNet16 model.

## Materials and methods

### Modeling methodology

**G-bneck principle.** The basic module of the GhostNet model, G-bneck, is shown in Fig 1.

The G-bneck module consists of two parts: one for expanding the number of channels and increasing the feature dimensions; and the other for reducing the number of channels to match the number of input channels. Batch normalization (BN) and activation function (ReLU) are applied after each layer, stride=1. In the case of stride=2, the Shortcut path is realized by a downsampling layer and a depthwise convolution (DwConv) with stride=2.

**Ghost module principle.** GhostNet is a lightweight network whose basic unit, Ghost Module, generates feature maps in two steps. The first step performs conventional convolution using a small number of convolution kernels to generate feature maps with fewer channels; the second step performs linear operations and convolution on the feature maps obtained in the first step to generate more features, which are then concatenated to obtain complete feature information. The principle of Ghost Module is shown in Fig 2a.

The specific calculations for Ghost Module are as follows:

$$Y' = Xf' \tag{1}$$

$$y_{i,j} = \Phi_{i,j}(y'_i) \forall i = 1, \dots, k \tag{2}$$

The first step is ordinary convolution, in Eq (1), is the input feature map, is the output original feature map, and is the convolution kernel used. The second step is to perform a linear operation on the original feature map output from the first step. In Eq (2), is the ith feature map in , is the jth linear mapping, and is the jth phantom feature map (ghost) generated from the ith original feature map. Finally, the feature map generated in the first step is connected (concat) with the feature map generated in the second step to obtain the complete

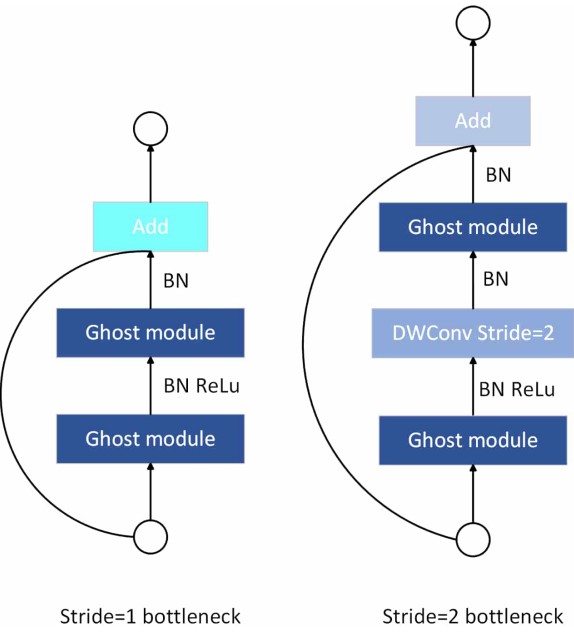

Stride=1 bottleneck          Stride=2 bottleneck

**Fig 1. G-bneck module.**

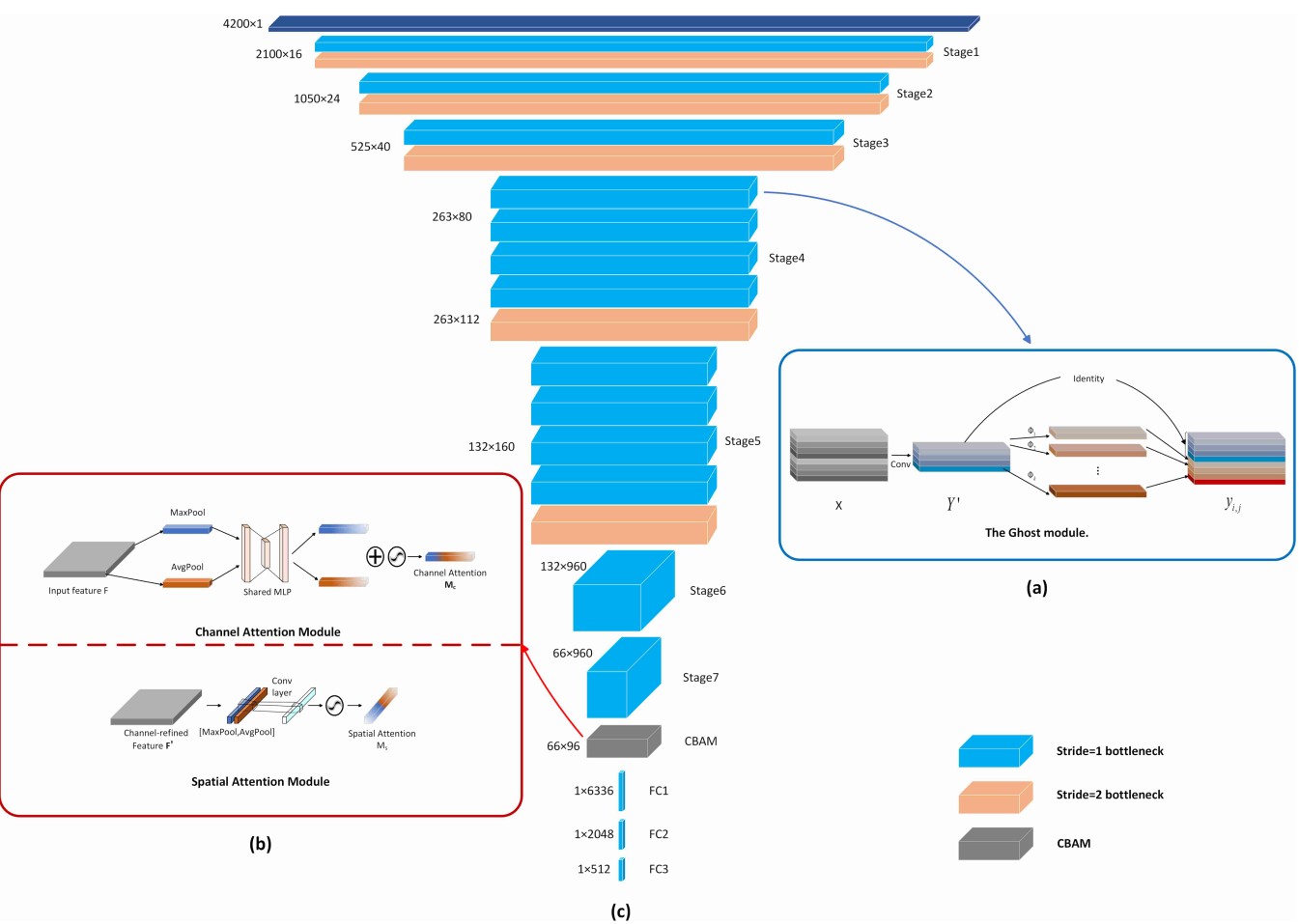

**Fig 2. Ghost module module.** a. Ghost module module b. Channel attention and spatial attention module c. GhostNet-CBAM network architecture. Where blue indicates the bottleneck layer for stride=1, orange indicates the bottleneck layer for stride=2, and gray indicates the dual attention module. The network as a whole consists of seven stages, CBAM modules, and three fully connected layers. Specific network parameters can be found in Table 1. GhostNet-CBAM network model parameters.

feature map. Compared to traditional convolutional operations, GhostNet extends the number of feature maps by generating additional feature maps by performing linear operations on fewer original feature maps. This approach greatly reduces the number of parameters and computational overhead of the model, making GhostNet more efficient while maintaining accuracy.

**CBAM principles.** The Convolutional Block Attention Module(CBAM) is a lightweight attention module capable of attending to both spatial and channel dimensions. The module consists of a channel attention module (CAM) and a spatial attention module (SAM). The channel attention module emphasizes regions of spectral data that are meaningful in different spaces, while the spatial attention module focuses on the location of spectral data features.

The channel attention mechanism is designed to adaptively adjust the weights of different channels to highlight the feature channels that are helpful for the current task. The features of the channel data are enhanced by global average pooling and global maximum

**Table 1. General network structure of GhostNet-CBAM.**

| Input | Operator | #exp | #out | #Stride |
|---|---|---|---|---|
| 4200 × 1 | Conv1d 3 × 3 | — | 16 | 2 |
| 2100 × 16 | G-bneck | 16 | 16 | 1 |
| 2100 × 16 | G-bneck | 48 | 24 | 2 |
| 1050 × 24 | G-bneck | 72 | 24 | 1 |
| 1050 × 24 | G-bneck | 72 | 40 | 2 |
| 525 × 40 | G-bneck | 120 | 40 | 2 |
| 525 × 40 | G-bneck | 240 | 80 | 2 |
| 263 × 80 | G-bneck | 200 | 80 | 1 |
| 263 × 80 | G-bneck | 184 | 80 | 1 |
| 263 × 80 | G-bneck | 184 | 80 | 1 |
| 263 × 80 | G-bneck | 480 | 112 | 1 |
| 263 × 112 | G-bneck | 672 | 112 | 1 |
| 263 × 112 | G-bneck | 672 | 160 | 2 |
| 132 × 160 | G-bneck | 960 | 160 | 1 |
| 132 × 160 | G-bneck | 960 | 160 | 1 |
| 132 × 160 | G-bneck | 960 | 160 | 1 |
| 132 × 160 | G-bneck | 960 | 160 | 1 |
| 132 × 160 | Conv1d 1 × 1 | — | 960 | 1 |
| 132 × 960 | AvgPool 7 × 7 | — | — | — |
| 66 × 960 | Conv1d 1 × 1 | — | 96 | 1 |
| 66 × 96 | CBAMLayer | 96 | 96 | — |
| 1 × 6336 | FC | — | 2048 | — |
| 1 × 2048 | FC | — | 512 | — |
| 1 × 512 | FC | — | 1 | — |

operator denotes core module. #exp denotes the expansion size. #out indicates the number of output channels.

pooling. This information is then summed and processed through a Sigmoid activation function to obtain the final channel attention weights. This module enhances the feature representation of important channels and suppresses minor channels, thus improving model performance.

The spatial attention mechanism aims to highlight the most relevant regions of the spectral data and reduce attention to less important regions. In the spatial attention mechanism, a spatial attention weight map is obtained by calculating the importance score of each spatial location through a convolutional layer or a separable convolutional layer. The weight map is multiplied channel-by-channel with the feature map to obtain a feature map weighted in the spatial dimension. In this way, the model can automatically select the spectral regions that are most distinguishable for the current task, improving the focus on important spatial locations. The operation steps are shown in Fig 2b.

**GhostNet-CBAM principle.** In this paper, the GhostNet model and the Convolutional Attention Module (CBAM) are first modified appropriately to enable it to handle one-dimensional data. Conv2d and BatchNorm2d of GhostModule, GhostBottleneck, and GhostNet in the GhostNet model are replaced using Conv1d and BatchNorm1d. GhostNet's AdaptiveAvgPool2d is not used for pooling and data compression, and MaxPool1d is used for pooling only to guarantee the integrity of data features. The final Linear layer is extended from the used one layer to three layers to fully guarantee the data features. For CBAM, change AdaptiveMaxPool2d and AdaptiveAvgPool2d to AdaptiveMaxPool1d and AdaptiveAvgPool1d in CBAMLayer. Then, based on the one-dimensional GhostNet model, a one-dimensional Convolutional Attention Module (CBAM) was added between the hidden and fully connected layers of the model, and the GhostNet-CBAM model was proposed for predicting soil pH.

The one-dimensional convolutional attention module is able to enhance both the channel information and the spatial information of the input data, and this dual-feature enhancement enables the GhostNet model to better capture the information about the pH parameters, thus enhancing the model's representational and perceptual capabilities and improving the prediction accuracy. The overall network architecture is shown in Fig 2c.

**Principles of the MCCV algorithm.** The Monte Carlo Cross Validation (MCCV) algorithm for the detection of abnormal samples is centered on the idea that there is a difference in the nature of normal and abnormal samples. When there are abnormal samples in the test set, it will have a greater impact on the prediction effect of the PLSR model. The MCCV method can screen out abnormal samples by building a large number of models and utilizing statistical parameters. The specific process is as follows: the MCCV first uses Monte Carlo sampling to randomly select the training and prediction sets from the original dataset, and builds a PLSR model based on the training set, and then makes predictions on the prediction set. The process is looped N times (usually $N \geq 1000$, in this paper $N = 1000$) to obtain a set of prediction residuals for the ith sample, denoted as $e_{i,j}(i = 1, 2, \ldots, m; j = 1, 2, \ldots, k)$, where k is the number of times a sample has been selected as the prediction set. The mean $\bar{e}_i$ and variance $s_i$ of the predicted residuals for each sample are calculated as follows:

$$\bar{e}_i = \frac{1}{k} \sum_{j=1}^{k} e_{ij} \tag{3}$$

$$s_i = \left( \frac{1}{k-1} \sum_{j=1}^{k} \left( e_{ij} - \bar{e}_i \right)^2 \right)^{\frac{1}{2}} \tag{4}$$

A mean-variance plot is drawn with the mean of the residuals as the horizontal coordinate and the variance of the residuals as the vertical coordinate. Sample points where the mean and variance deviate significantly from the main body are considered abnormal samples.

According to the standardization formula, the standardized residual mean and standardized residual variance corresponding to $\bar{e}_i$ and $s_i$ are calculated as follows:

$$\delta_{\bar{e}_i} = \frac{\bar{e}_i - mean(\bar{e}_i)}{std(\bar{e}_i)} \tag{5}$$

$$\delta_{s_i} = \frac{s_i - mean(s_i)}{std(s_i)} \tag{6}$$

where $mean\left(\bar{e}_i\right)$ and $std\left(\bar{e}_i\right)$is the standard deviation of all sample residual means; $mean\left(s_i\right)$is the mean of all sample residual variances and $std\left(s_i\right)$ is the standard deviation of all sample residual variances. where $\delta_{\bar{e}_i}$ is the mean of all sample residual means and $\delta_{s_i}$ is the standard deviation of all sample residual means;When $\delta_{\bar{e}_i}$and $\delta_{s_i}$ are equal to 2 is the threshold value of $\bar{e}_i$ and $s_i$ which is calculated as:

$$T_m = mean\left(\bar{e}_i\right) + 2std\left(\bar{e}_i\right) \tag{7}$$

$$T_s = mean\left(s_i\right) + 2std\left(s_i\right) \tag{8}$$

$T_m$ and $T_s$ are the thresholds for the residual mean and the residual variance, respectively. If the $\bar{e}_i$ and $s_i$of a sample exceed the threshold $T_m$ or $T_s$, it is considered an abnormal sample.

Improved Monte Carlo sampling to optimize the MCCV algorithm starts with the assumption that there are anomalous samples in the initial build sample. Before building the prediction model, the samples with smaller standardized residuals (less than the threshold) were selected as the modeling subset by the MCCV algorithm. From this, 80% is selected as the training set and the remaining 20% and the samples with large normalized residuals are used as the test set. This not only increases the difference between the predicted residuals of the abnormal and normal samples, but also reduces the masking effect of the outliers. Repeat the steps of the MCCV algorithm N times and calculate and for each sample. If and deviate significantly from the subject, the sample is considered abnormal.

## LUCAS soil spectral library

Soil data for this study were obtained from the LUCAS 2015 topsoil database provided by the European Soil Data Center (ESDAC). The database collects samples from 27,069 geo-referenced points in 28 countries selected on the basis of land-use and topographic information. The sample sites selected were classified by FAO to include leached soils, submerged soils, initiated soils, black soils, ashed soils, and anthropogenic soils. Spectral data were collected by rotating the sample container and collecting two spectra per reference point to capture different regions of the soil sample. Spectral data were measured using a FOSS XDS fast analyzer with a band range of 400 nm to 2499.5 nm and an optical interval of 0.5 nm for a total of 4200 bands of absorbance spectra. After removing invalid samples due to unrecognizability, mislabeling, and mismanagement, 21,778 each of the two directions of data were ultimately retained. In the following, $S$ and $P$ will be used to represent the soil spectral data obtained from measurements in different directions, respectively. All samples were analyzed in an ISO-certified laboratory using standard test methods for the determination of 12 physicochemical properties of topsoil, such as organic carbon ($OC$) content, nitrogen ($N$) content, phosphorus ($P$) content, potassium ($K$) content and pH($H_2O$ and $CaCl_2$). This paper focuses on the pH of the soil. The reference values for pH in the data were obtained through two standards, $H_2O$ and $CaCl_2$, so there are two sets of spectral data ($S$ and $P$ directions) and two reference values in this paper, for a total of four datasets, called: $S\_H_2O$, $S\_CaCl_2$, $P\_H_2O$, and $P\_CaCl_2$.

## Model evaluation

In the regression model, the coefficient of determination ($R^2$) and root mean square error (RMSE) were used to assess the goodness of fit and accuracy of the model. The closer the value of $R^2$ is to 1 and the closer the value of RMSE is to 0, the better the predictive performance and stability of the model.

$$R^2 = 1 - \frac{\sum_{i=1}^{n} (y_i - \hat{y}_i)^2}{\sum_{i=1}^{n} (y_i - \bar{y}_i)^2} \tag{9}$$

$$RMSE = \sqrt{\frac{1}{n} \sum_{i=1}^{n} (y_i - \hat{y}_i)^2} \tag{10}$$

where $n$ is the number of samples, $y_i$ is the measured pH value, $\hat{y}_i$ is the pH value predicted by the model, and $\bar{y}_i$ is the average pH value measured.

## Model training

The main hyperparameters for training the model are as follows: the batch size is (batch-size) is 16. The learning rate is 0.0001, the activation function is Relu, the optimizer is Adam, and the epochs are 300. The model was realized using Pytorch 1.13.1. Hardware environment: CPU is Intel Core i5-12490F 4.60 GHz, RAM is 32 GB; GPU is NVIDIA 3060Ti.

The number of parameters obtained from the training of VGGNet16, GhostNet, and GhostNet-CBAM are 1130.78 Mb, 58.43 Mb, and 58.49 Mb, respectively, and the number of samples that can be processed per second are 1635, 3168, and 3100.

# Results

## Statistical analysis of soil pH

Table 2 Statistical data describes the basic information about the datasets used in this paper. $H_2O$ and $CaCl_2$ in the dataset names represent the pH values obtained from measurements at different solutions, respectively. The dataset starting with Re refers to the dataset after the MCCV algorithm removes the anomalous samples.

The statistics showed that there were six samples with pH values greater than 8.30 and 9.07 in both the pH($H_2O$) and pH($CaCl_2$) datasets, and the exclusion of these samples had a small impact on the overall pH parameter predictions. The median pH value in the dataset is similar to the mean, indicating that the data are normally distributed. The standard deviation for all data sets ranged from 1.3305 to 1.3992, which is lower than the mean, indicating less variation in soil pH. In addition, the skewness coefficients of the pH($ReS\_H_2O$) and pH($ReP\_H_2O$) data were 0.0027 and 0.0032, respectively, indicating an essentially symmetric distribution, while the skewness coefficients of the pH($S\_CaCl_2$), pH($P\_CaCl_2$), pH($ReS\_CaCl_2$) and pH($ReP\_CaCl_2$) data were 0.1363, 0.1363, 0.1934, and 0.1991, indicating that the distributions are somewhat right skewed. The skewness coefficients for both pH($S\_H_2O$) and pH($P\_H_2O$) data were –0.1173, indicating some leftward bias in the distribution. The specific spectral profile is shown in Fig 3.

## CBAM's role in modeling

Two models, GhostNet and GhostNet-CBAM, were used to regressively predict the pH parameters of the soil, and the accuracy of the models on the test set was compared using $R^2$ and RMSE (see Table 3). In predicting soil properties, the prediction accuracy of GhostNet-CBAM was significantly better than that of GhostNet on the $S\_H_2O$, $P\_H_2O$ and $P\_CaCl_2$ datasets. However, the prediction ability of GhostNet-CBAM on the $S\_CaCl_2$ dataset is slightly lower than that of GhostNet. GhostNet-CBAM gave the best prediction on the

**Table 2. Statistics on basic soil information.**

| Data | Size | Min | Max | Med | AVE | STD | Skewness |
|---|---|---|---|---|---|---|---|
| $S\_H_2O$ | 21778 | 2.6000 | 10.0000 | 5.8000 | 5.7453 | 1.3992 | –0.1173 |
| $S\_CaCl_2$ | 21778 | 3.1700 | 10.3700 | 6.0700 | 6.1312 | 1.3461 | 0.1363 |
| $P\_H_2O$ | 21778 | 2.6000 | 10.0000 | 5.8000 | 5.7453 | 1.3992 | –0.1173 |
| $P\_CaCl_2$ | 21778 | 3.1700 | 10.3700 | 6.0700 | 6.1312 | 1.3461 | 0.1363 |
| $ReS\_H_2O$ | 19924 | 2.6000 | 8.3000 | 5.7000 | 5.7013 | 1.3923 | 0.0027 |
| $ReS\_CaCl_2$ | 19896 | 3.1700 | 9.0700 | 6.0000 | 6.0858 | 1.3305 | 0.1934 |
| $ReP\_H_2O$ | 19962 | 2.6000 | 8.3000 | 5.7000 | 5.7015 | 1.3925 | 0.0032 |
| $ReP\_CaCl_2$ | 19951 | 3.1700 | 9.0700 | 6.0000 | 6.0883 | 1.3295 | 0.1991 |

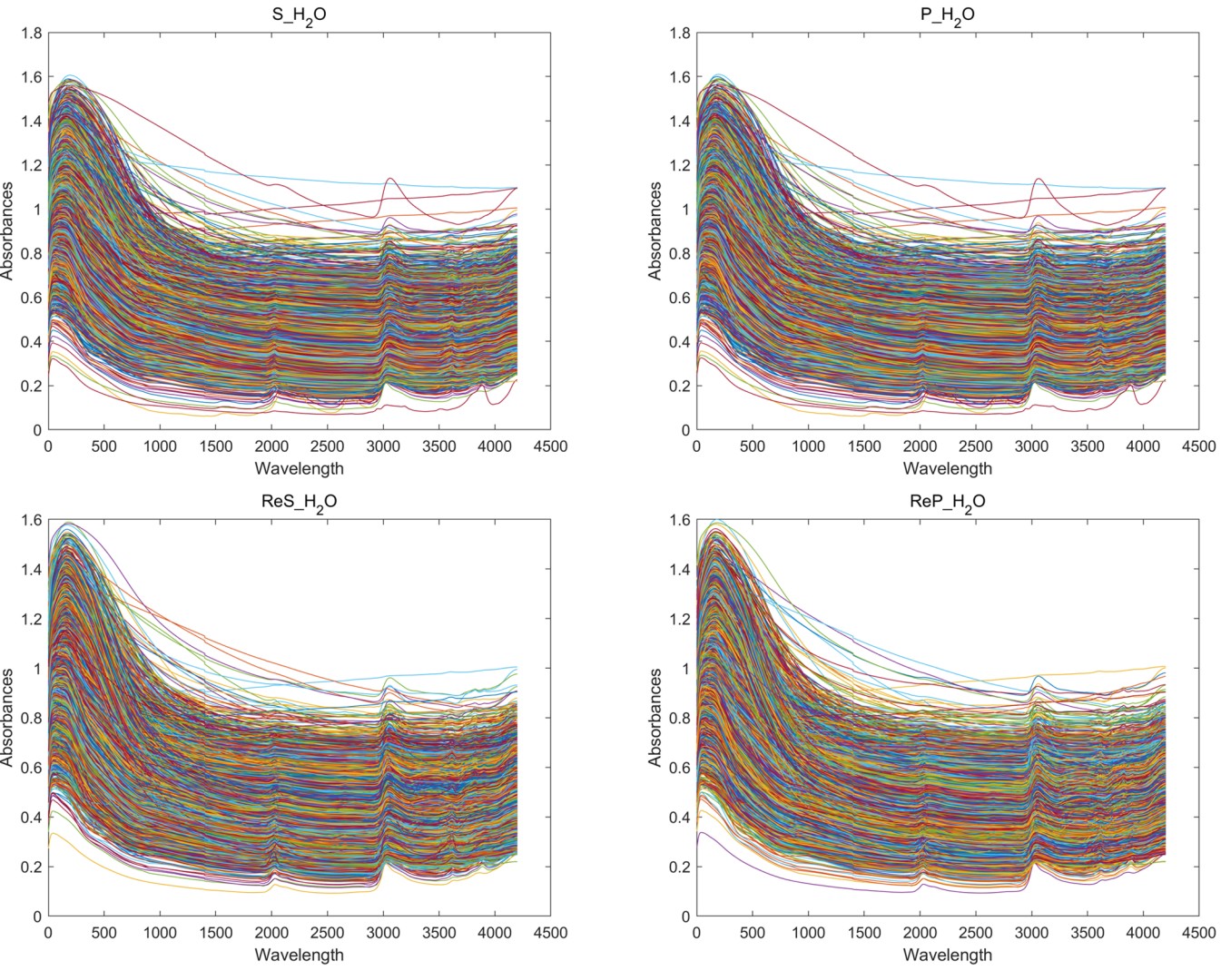

**Fig 3. Spectrograms of data from different datasets.**

**Table 3. GhostNet-CBAM and GhostNet model prediction results.**

| Model | Data | RMSE | R2 |
|---|---|---|---|
| GhostNet | $S\_H_2O$ | 0.4727 | 0.8821 |
| | $S\_CaCl_2$ | 0.4445 | 0.8889 |
| | $P\_H_2O$ | 0.3923 | 0.9198 |
| | $P\_CaCl_2$ | 0.3876 | 0.9158 |
| GhostNet-CBAM | $S\_H_2O$ | 0.4351 | 0.9001 |
| | $S\_CaCl_2$ | 0.4451 | 0.8886 |
| | $P\_H_2O$ | 0.3681 | 0.9294 |
| | $P\_CaCl_2$ | 0.3776 | 0.9201 |

$P\_H_2O$ dataset with an $R^2$ of 0.9294 and an RMSE of 0.3681. In summary, the GhostNet model incorporating the CBAM performs better in terms of goodness-of-fit and predictio accuracy.

## The role of the MCCV algorithm for prediction

In this study, the pH($P\_H_2O$) dataset was first processed using the MCCV algorithm. Subsequently, the total mean $\bar{e}_i$ and variance $s_i$ of the residuals were calculated and the thresholds $T_m$ were set to 1.0015 and $T_s$ to 0.0893 based on the thresholding formula. A mean-variance plot was plotted as shown in Fig 4a, using the mean of the residuals as the horizontal coordinate and the variance of the residuals as the vertical coordinate.

The data with small normalized residuals in Fig 4a is selected as the training set, and the rest of the data and the data outside the threshold are used as the test set for PLSR modeling. The prediction of the prediction set and the data with large standardized residuals is performed to obtain the residual values. Repeating the above steps 1000 times, the mean value of the residuals was calculated to be 0.9587 and the variance of the residuals to be 0.0874, plotting the mean-variance plot shown in Fig 4b.

The scatter samples exceeding the threshold in Fig 4 are considered as abnormal samples. For the pH($P\_H_2O$) dataset, a total of 1816 samples are anomalous samples, and 19,962 normal samples remain after excluding the anomalous samples.

The anomalous data in the $S\_H_2O$ $S\_CaCl_2$ $P\_H_2O$ and $P\_CaCl_2$ datasets are rejected using the above steps. The data before and after culling were modeled using the GhostNet model to predict their pH parameters. Table 4 demonstrates the overall prediction accuracy of the model for the four data sets. From the table, it can be seen that the prediction accuracy of GhostNet is significantly better than the prediction accuracy using the original data when using the data culled by the MCCV algorithm as the training and test sets. RMSE has a maximum boost of 0.1107 in the $S\_H_2O$ dataset, and $R^2$ has a maximum boost of 0.0485 in the $S\_H_2O$ dataset. This suggests that model training on the culled dataset can significantly improve the prediction of parameters.

## Comparison of GhostNet-CBAM with other models

Regression prediction was performed on four datasets using three different models(PLSR, VGGNet-16, GhostNet-CBAM). Table 5 demonstrates the overall prediction accuracy for the test sets. In these four datasets, the prediction accuracy of GhostNet-CBAM is better than that

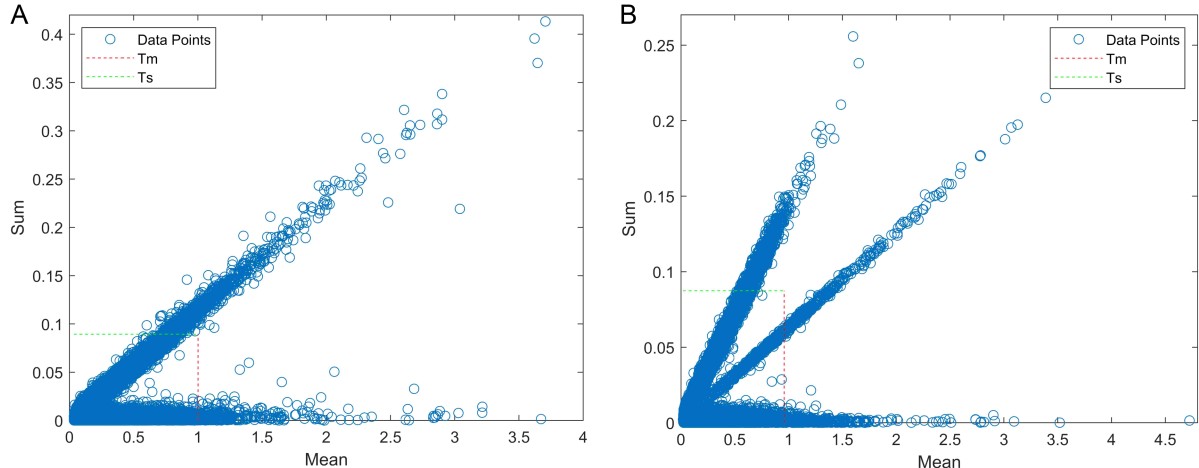

**Fig 4. Graph of the results of the MCCV algorithm.**

**Table 4. Prediction results for original and optimized data.**

| Model | Data | RMSE | R2 |
|---|---|---|---|
| Original | $S\_H_2O$ | 0.4603 | 0.8882 |
| | $S\_CaCl_2$ | 0.4451 | 0.8886 |
| | $P\_H_2O$ | 0.4044 | 0.9148 |
| | $P\_CaCl_2$ | 0.3813 | 0.9185 |
| MCCV | $S\_H_2O$ | 0.3496 | 0.9367 |
| | $S\_CaCl_2$ | 0.3719 | 0.9228 |
| | $P\_H_2O$ | 0.3267 | 0.9447 |
| | $P\_CaCl_2$ | 0.3544 | 0.9291 |

**Table 5. PLSR, VGGNet16, and GhostNet-CBAM prediction results.**

| Model | Data | RMSE | R2 |
|---|---|---|---|
| PLSR | $S\_H_2O$ | 0.4348 | 0.9016 |
| | $S\_CaCl_2$ | 0.4285 | 0.8956 |
| | $P\_H_2O$ | 0.4184 | 0.9088 |
| | $P\_CaCl_2$ | 0.4265 | 0.8969 |
| VGGNet16 | $S\_H_2O$ | 0.3834 | 0.9239 |
| | $S\_CaCl_2$ | 0.3814 | 0.9188 |
| | $P\_H_2O$ | 0.3473 | 0.9375 |
| | $P\_CaCl_2$ | 0.3665 | 0.9242 |
| Ghost-CBAM | $S\_H_2O$ | 0.3496 | 0.9367 |
| | $S\_CaCl_2$ | 0.3719 | 0.9228 |
| | $P\_H_2O$ | 0.3267 | 0.9447 |
| | $P\_CaCl_2$ | 0.3544 | 0.9291 |

of VGGNet-16 and PLSR. Compared with PLSR, GhostNet-CBAM has a significant advantage in RMSE metrics, with RMSE($P\_H_2O$) decreasing from 0.4184 to 0.3267, a difference of 0.0917, and the model fit goodness of fit improving by 3.59%. Compared with VGGNet-16, GhostNet-CBAM reduces the RMSE ($S\_H_2O$) by up to 0.0203 and improves the model fit goodness by up to 1.28%. It can be seen that the GhostNet-CBAM model proposed in this paper has a better effect on the regression prediction of soil pH parameters.

In this paper, the overall distribution of the predicted values of different models for each soil data and their reference values are analyzed through box-and-line plots, as shown in Fig 5.

Fig 5 shows the statistics of the reference values of the test set and the predictions of the three models for the test set. The horizontal coordinates indicate the reference and predicted values of the different models and the vertical coordinates indicate the pH values. The figure caption specifies the dataset used, where T of GhostCBAM-T denotes the reference value of the test set and P of GhostCBAM-P denotes the predicted value of the test set. From the figure, it can be seen that the maximum and minimum values of the prediction results of the PLSR model on the four datasets are significantly different from the reference values, especially the prediction errors on the $S\_H_2O$ dataset are more obvious, with the maximum and minimum values of the reference values of 2.7 and 8.1, respectively, while the minimum and maximum values of its prediction results are 1.800 and 10.3053, respectively. In addition, the upper and lower quartiles of the PLSR model predictions differed from the reference value by an average of 0.1765, indicating that the PLSR model has poor prediction accuracy. In contrast, the maximum, minimum and quartile points of the prediction results of the GhostNet-CBAM and VGGNet16 models are closer to the reference values. However, the median prediction of the GhostNet-CBAM model is closer to the reference value than that of the VGGNet16 model, and the statistics show that the GhostNet-CBAM median error

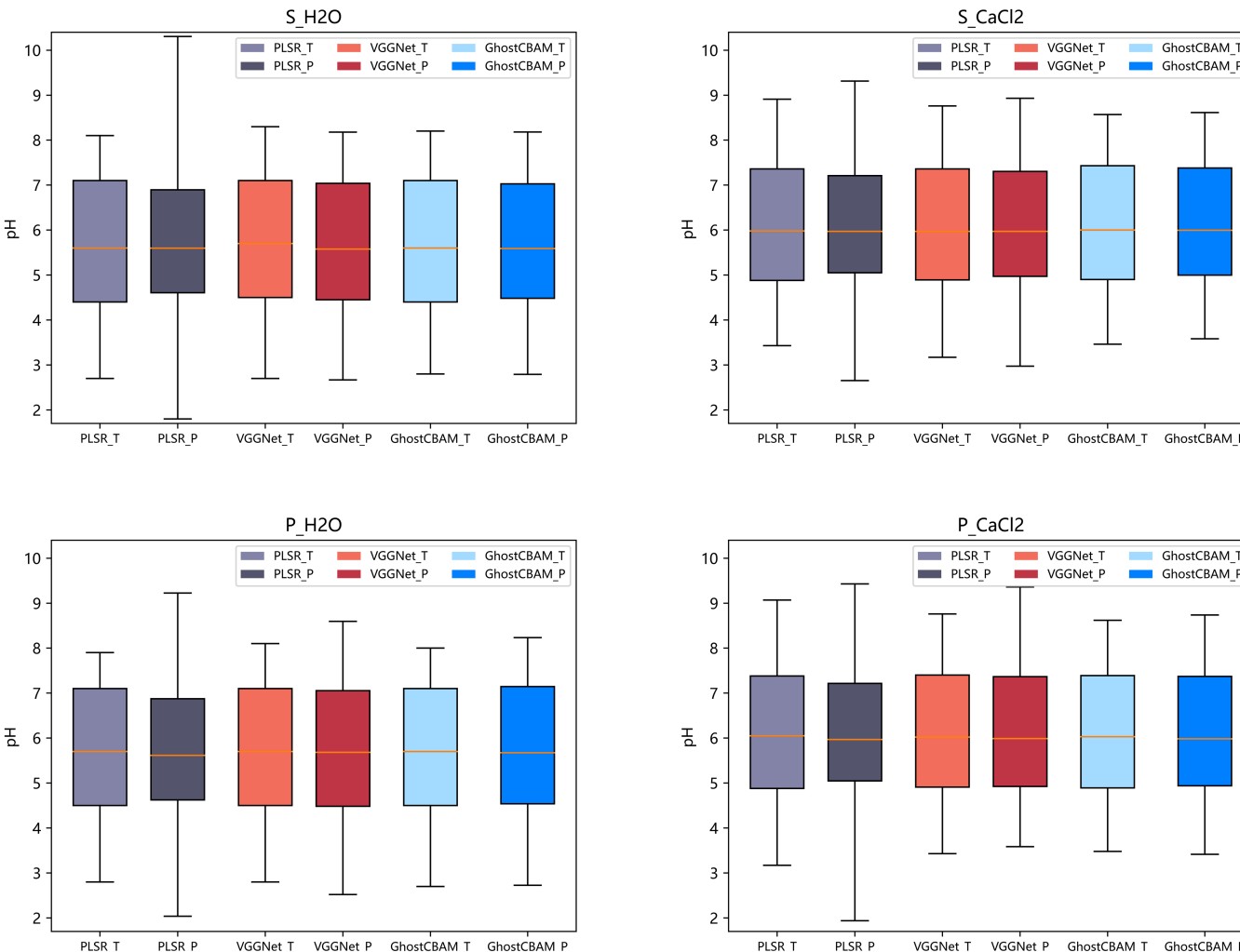

**Fig 5. Boxplot of model predictions versus reference values.**

improved by 47.2% (from 0.04322 to 0.0228). Combining the RMSE and $R^2$ parameters, it can be seen that the predictive ability of the GhostNet-CBAM model is better than that of the VGGNet16 model.

In order to fully analyze the predictive ability of different models, this paper plots the reference values and predictions of different datasets into a scatter density plot, as shown in Fig 6.

The above figure shows the scatter density plots of the predictions of the three models PLSR, VGGNet16 and GhostNet-CBAM for the dataset filtered by the MCCV algorithm. Each row in the figure represents a dataset and each column represents a model. The horizontal coordinate represents the true pH of the sample and the vertical coordinate represents the pH predicted by the model. The black dashed line indicates the optimal prediction result, i.e., the closer the scatter is to the black dashed line, the better the model's prediction is. A least squares fit to the predictions of each model yields the red diagonal line. The semi-transparent red shaded area in the figure represents the probability that the difference between the prediction result and the true value is less than 0.3. The smaller the shaded area is, the

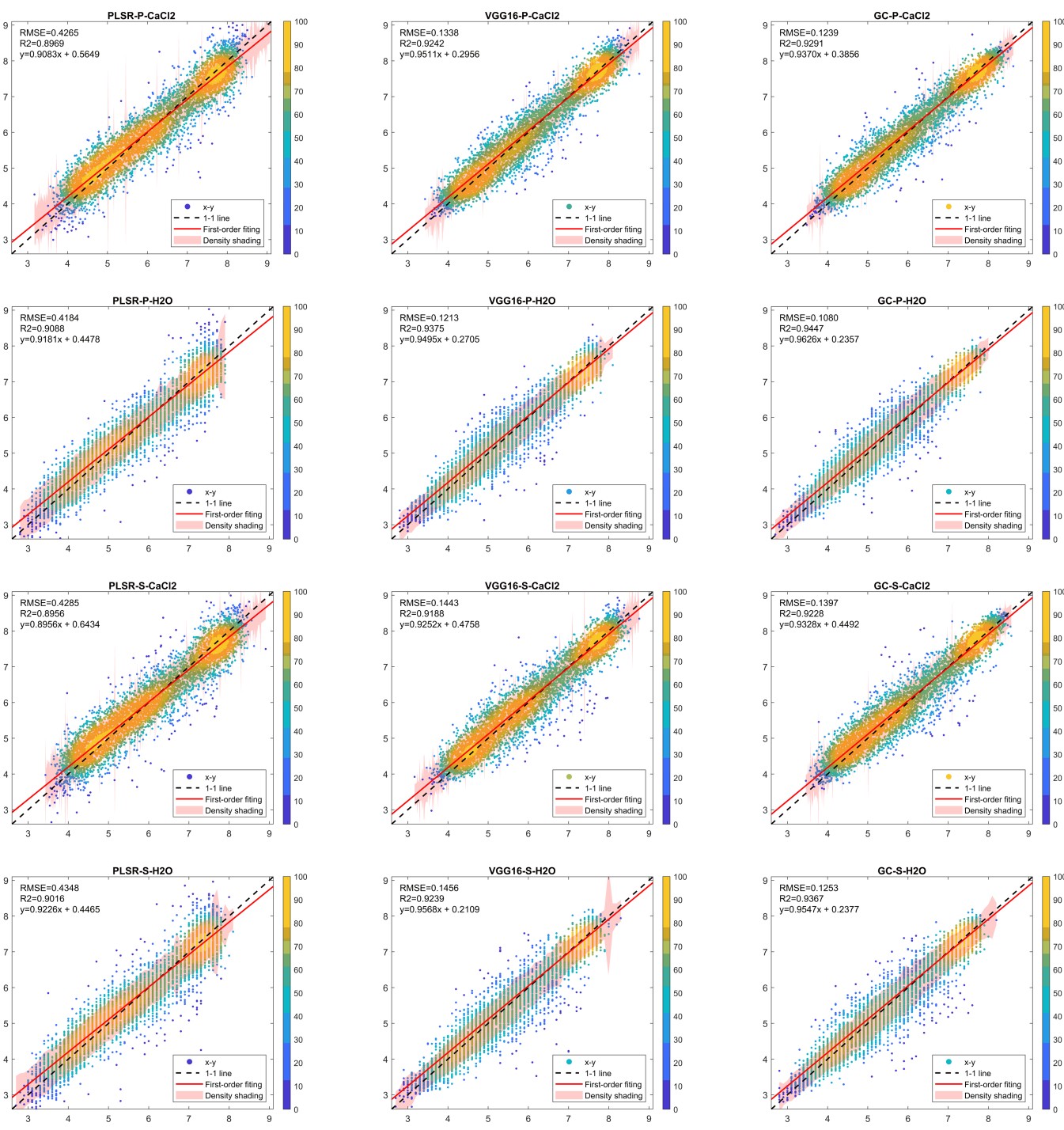

**Fig 6. Scatter density plot of reference values vs. predictions.**

greater the density of the area falling in the qualified area, i.e., the prediction result is more accurate.

The RMSE, $R^2$, and the first-order fit line for this prediction are labeled in the upper left corner of the figure. As can be seen from the figure, the PLSR model is weaker than the

GhostNet-CBAM model in terms of the degree of scatter aggregation, RMSE and $R^2$ values, and the first-order fit line. Comparison between the prediction results of VGGNet16 and the GhostNet-CBAM model showed that the first-order fit line of the GhostNet-CBAM model was slightly inferior only in the $P\_CaCl_2$ dataset, and all the other datasets showed some improvement in each of the metrics. As can be seen from the size of the red shaded area, the shaded area of the VGGNet16 and GhostNet-CBAM models is basically smaller than that of the PLSR model in the whole pH interval, indicating that the prediction results of the VGGNet16 and GhostNet-CBAM models are closer to the reference value and have better prediction ability. Comparing the red shaded areas of the VGGNet16 and GhostNet-CBAM models, the red area of GhostNet-CBAM is mostly smaller than that of VGGNet16 in the intervals of 2.6 to 4.9 and 6.8 to 8.1, whereas in the interval of 5.0 to 6.7, the areas of the two are basically the same. Taking $P\_H_2O$ as an example, the normalization was calculated to be 0.5749, 0.4493 and 0.4294 for PLSR, VGGNet16 and GhostNet-CBAM unit areas, respectively. This indicates that the GhostNet-CBAM model has overall better predictive ability than the VGGNet16 model, and the predictive advantage is more pronounced in the intervals of 2.6 to 4.9 and 6.8 to 8.1.

## Conclusion

In this paper, a combined algorithm of GhostNet combined with CBAM was used to predict pH parameters in soil. GhostNet, as a lightweight neural network model, has fewer parameters to shorten the training time and improve the prediction efficiency, the CBAM module highlights the key information and enhances the feature extraction and parameter prediction ability by assigning different weights to the hidden layer units, and the MCCV, as a preprocessing algorithm, can eliminate the abnormal samples in the original data to further optimize the data and improve the prediction accuracy. From the $R^2$ and RMSE parameters, as well as the scatter density plots, it was confirmed that the method can quickly and accurately obtain the PH parameters of the soil. Future research in this field will be directed towards the combination of spectral and remote sensing data for the integrated evaluation of intrinsic and extrinsic qualities of soil parameters [23].

## Author contributions

**Conceptualization:** Wenjin Wang, Peng Tian.

**Data curation:** Jianguo Zhu.

**Formal analysis:** Jianguo Zhu.

**Methodology:** Peng Tian.

**Software:** Jianguo Zhu.

**Writing – original draft:** Jianguo Zhu.

**Writing – review & editing:** Wenjin Wang, Peng Tian.

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
