## [Decision Letter · Decision Letter 0]

PONE-D-24-54324Spectroscopic measurement of near-infrared soil pH parameters based on GhostNet-CBAMPLOS ONE

Dear Dr. Tian,

Thank you for submitting your manuscript to PLOS ONE. After careful consideration, we feel that it has merit but does not fully meet PLOS ONE’s publication criteria as it currently stands. Therefore, we invite you to submit a revised version of the manuscript that addresses the points raised during the review process.

We look forward to receiving your revised manuscript.

Kind regards,

Himadri Majumder, Ph.D

Academic Editor

PLOS ONE

Journal Requirements:

“This research was funded by the Key Research Foundation of Hunan Province Education Department (No.21A0404)”

Reviewers' comments:

Reviewer's Responses to Questions

**Comments to the Author**

1. Is the manuscript technically sound, and do the data support the conclusions?

Reviewer #1: Partly

Reviewer #2: Yes

2. Has the statistical analysis been performed appropriately and rigorously? 

Reviewer #1: No

Reviewer #2: Yes

3. Have the authors made all data underlying the findings in their manuscript fully available?

Reviewer #1: No

Reviewer #2: Yes

4. Is the manuscript presented in an intelligible fashion and written in standard English?

Reviewer #1: Yes

Reviewer #2: Yes

5. Review Comments to the Author

Reviewer #1: 1.Considering that the paper primarily uses publicly available datasets for experiments and employs algorithms that are not newly proposed, while focusing more on deep learning model aspects and neglecting soil science perspectives, the paper lacks innovation and mechanistic depth. Additionally, the feasibility of the research methods requires further validation.

2.It is recommended to supplement the introduction with the specific principles of using near-infrared spectroscopy to estimate soil pH values, along with a review of related research progress.

3.Since GhostNet is an important innovation point of the paper, it is suggested to include relevant literature on this method in the introduction to clarify the scenarios in which GhostNet is more applicable.

4.Although the paper uses data from publicly available datasets, it should still provide details about the experimental conditions, soil types, and spectral curve characteristics of the adopted data.

5.From the results, the methods employed in the paper do not appear to significantly outperform VGG-16. It is recommended to conduct comparative experiments with methods used in soil pH detection studies published over the past three years.

6.The conclusion of the results section only analyzes the model's performance in terms of accuracy for soil pH detection, without incorporating relevant soil science aspects. For instance, the paper should discuss whether factors such as soil types, soil composition ratios, and the spectral characteristics of different components have impacted the research results.

Reviewer #2: 1. It is necessary to enhance the description of the latest research findings in the introduction.

2. What problem does the innovation of this study primarily aim to solve? A review of the current research deficiencies is needed.

3. The title of Table 2 needs to be revised.

4. It is recommended to add 1-2 additional comparison algorithms.

5. It is necessary to specify the method used to process the raw spectral data.

6. The title of Figure 2 also needs to be revised.

7. The conclusion section should present several issues that require further research.

8. The following article is worth referring to: https://doi.org/10.1007/s11694-024-02787-1.

6. PLOS authors have the option to publish the peer review history of their article (what does this mean?). If published, this will include your full peer review and any attached files.

Reviewer #1: No

Reviewer #2: **Yes: **Hongbiao Zhou

---

## [Author Response · Author response to Decision Letter 1]

24 Apr 2025

We greatly appreciate the comments by reviewers which helped us enhance the manuscript significantly in the revised version. The point-to-point responses are shown below in blue with related changes implemented in the revised manuscript.

Reviewer#1

Q1. Considering that the paper primarily uses publicly available datasets for experiments and employs algorithms that are not newly proposed, while focusing more on deep learning model aspects and neglecting soil science perspectives, the paper lacks innovation and mechanistic depth. Additionally, the feasibility of the research methods requires further validation.

Thank you for your suggestions and feedback. The dataset used in this paper is the soil spectral data published by ESDAC for the year 2015, and the GhostNet model was at first mainly used for the classification of images (when was GhostNet proposed and what was it mainly used for). However, we are not aware of anyone using the GhostNet model to process LUCAS data for soil parameter prediction. We focused on the balance between model efficiency and accuracy, so we compared the GhostNet model with the original models (PLSR , VGGNet). We used 80% of the LUCAS dataset as training data and the remaining 20% as test data, and the predicted results obtained were compared with the real pH values, and the results are shown in Figure 6 of the paper as GC-P-H2O. We find that the predicted results are in better agreement with the real pH and compare this result with PLSR as well as VGGNet16 model. The predicted shaded area per unit is 0.5749, 0.4493 and 0.4294, respectively, and the smaller the shaded area, the better the prediction ability of the model. From the comparison results, it can be seen that the GhostNet-CBAM model has an advantage over the other models.

Q2. It is recommended to supplement the introduction with the specific principles of using near-infrared spectroscopy to estimate soil pH values, along with a review of related research progress.

Thank you for your suggestions and feedback. The first paragraph of the introduction describes the specific principles of soil pH estimation by near infrared spectroscopy (VNIR). The main principle is that VNIR spectra are the result of radiation absorption by various chemical bonds (e.g., C-H, N-H, and O-H) in soil samples. pH parameters can be quantitatively determined by correlating VNIR spectra with the main spectrally active components.

We have added the latest research progress to the original relevant research progress[1][2].

[1] Zang D, Zhao Y, Luo C, et al. Improving the accuracy of soil organic matter mapping in typical Planosol areas based on prior knowledge and probability hybrid model[J]. Soil and Tillage Research, 2025, 246: 106358.

[2] Wang X, Zhang M W, Zhou Y N, et al. Simultaneous estimation of multiple soil properties from vis-NIR spectra using a multi-gate mixture-of-experts with data augmentation[J]. Geoderma, 2025, 453: 117127.

Q3. Since GhostNet is an important innovation point of the paper, it is suggested to include relevant literature on this method in the introduction to clarify the scenarios in which GhostNet is more applicable.

Thank you for your suggestions and feedback. The first paragraph of the introduction describes the specific principles of soil pH estimation by near infrared spectroscopy (VNIR)[3]. The main principle is that VNIR spectra are the result of radiation absorption by various chemical bonds (e.g., C-H, N-H, and O-H) in soil samples. pH parameters can be quantitatively determined by correlating VNIR spectra with the main spectrally active components.

[3] Paoletti M E, Haut J M, Pereira N S, et al. Ghostnet for hyperspectral image classification[J]. IEEE Transactions on Geoscience and Remote Sensing, 2021, 59(12): 10378-10393.

Q4. Although the paper uses data from publicly available datasets, it should still provide details about the experimental conditions, soil types, and spectral curve characteristics of the adopted data.

Thank you for your suggestions and feedback. We have added a description of the FAO classification of the soils in the “LUCAS soil spectral library” subsection in red font. Figure 3 has been added to show the spectral profiles specific to the dataset used in this paper.

Q5. From the results, the methods employed in the paper do not appear to significantly outperform VGG-16. It is recommended to conduct comparative experiments with methods used in soil pH detection studies published over the past three years.

Thank you for your suggestions and feedback. Due to time constraints and the lack of a specific published code on the web, we only compare the experimental results with those of published papers. A closer comparison with the results in [2][4] is shown below.

Model RMSE R2

RPK 0.5900 0.8100

MMoE 0.3800 0.9200

GhostNet-CBAM 0.3267 0.9447

From the comparison results, it is found that the GhostNet-CBAM model has some advantages in both RMSE and R2 parameter indicators.

[2] Wang X, Zhang M W, Zhou Y N, et al. Simultaneous estimation of multiple soil properties from vis-NIR spectra using a multi-gate mixture-of-experts with data augmentation[J]. Geoderma, 2025, 453: 117127.

[4] Xiao S, Ou M, Geng Y, et al. Mapping soil pH levels across Europe: An analysis of LUCAS topsoil data using random forest kriging (RFK)[J]. Soil Use and Management, 2023, 39(2): 900-916.

Q6. The conclusion of the results section only analyzes the model's performance in terms of accuracy for soil pH detection, without incorporating relevant soil science aspects. For instance, the paper should discuss whether factors such as soil types, soil composition ratios, and the spectral characteristics of different components have impacted the research results.

Thank you for your suggestions and feedback. The focus of our dissertation discussion is to predict soil pH parameters using the GhostNet-CBAM model with a certain equilibrium of efficiency and accuracy of prediction. The effect of soil characteristics on pH prediction was not discussed in the thesis due to the need to modify the GhostNet model as well as incorporate the attention mechanism and compare it with other models. The conclusions of the reviewers on the effects of other soil characteristics on pH parameters can be found in some of the following papers [4][5][6].

[4] Xiao S, Ou M, Geng Y, et al. Mapping soil pH levels across Europe: An analysis of LUCAS topsoil data using random forest kriging (RFK)[J]. Soil Use and Management, 2023, 39(2): 900-916.

[5] Mosley L M, Rengasamy P, Fitzpatrick R. Soil pH: Techniques, challenges and insights from a global dataset[J]. European Journal of Soil Science, 2024, 75(6): e70021.

[6] Miller R O, Kissel D E. Comparison of soil pH methods on soils of North America[J]. Soil Science Society of America Journal, 2010, 74(1): 310-316.

Reviewer#2

Q1. It is necessary to enhance the description of the latest research findings in the introduction.

Thank you for your suggestions and feedback. We have added two recent articles in the fourth paragraph of the introduction to enhance the description of recent research progress. The specific literature is [1][2].

Wang et al. proposed a model based on multi-gate hybrid expert network (MMoE), which can improve the prediction accuracy of soil parameters by combining with data enhancement. Compared with the traditional single- and multi-task models, MMoE reduces the RMSE by 5\%-48\% and improves the R2 by 1\%-119\%. Li et al. proposed a lightweight deep learning model, DSCformer, to predict soil parameters, which combines the Metaformer architecture with deep separable convolution to reduce the computational complexity and improve the accuracy, and analyzes the interpretability with the help of SHAP method, which specifies the spectral bands related to soil nitrogen content. It analyzes the interpretability with the help of SHAP method and clarifies the spectral bands related to soil nitrogen content.

[1] Zang D, Zhao Y, Luo C, et al. Improving the accuracy of soil organic matter mapping in typical Planosol areas based on prior knowledge and probability hybrid model[J]. Soil and Tillage Research, 2025, 246: 106358.

[2] Wang X, Zhang M W, Zhou Y N, et al. Simultaneous estimation of multiple soil properties from vis-NIR spectra using a multi-gate mixture-of-experts with data augmentation[J]. Geoderma, 2025, 453: 117127.

Q2. What problem does the innovation of this study primarily aim to solve? A review of the current research deficiencies is needed.

Thank you for your suggestions and feedback. Most of the existing machine learning models have low prediction accuracy in predicting soil parameters after modeling from large spectral data. This may be due to the fact that most machine learning models are difficult to extract the features of high-dimensional complex spectral data.The parameters of the VGGNet16 model are too large, and its model size is larger than 1 GB, which is not favorable for the practical application of the model. The aim of this paper is to establish a soil pH parameter prediction model with a smaller model and better prediction accuracy.

Q3. The title of Table 2 needs to be revised.

Thank you for your suggestions and feedback. This was a problem caused by an oversight on our part, and it has been changed to “Statistics on basic soil information.”

Q4. It is recommended to add 1-2 additional comparison algorithms.

Thank you for your suggestions and feedback. Due to time constraints and the lack of a specific published code on the web, we only compare the experimental results with those of published papers. A closer comparison with the results in [2][4] is shown below.

Model RMSE R2

RPK 0.5900 0.8100

MMoE 0.3800 0.9200

GhostNet-CBAM 0.3267 0.9447

From the comparison results, it is found that the GhostNet-CBAM model has some advantages in both RMSE and R2 parameter indicators.

[2] Wang X, Zhang M W, Zhou Y N, et al. Simultaneous estimation of multiple soil properties from vis-NIR spectra using a multi-gate mixture-of-experts with data augmentation[J]. Geoderma, 2025, 453: 117127.

[4] Xiao S, Ou M, Geng Y, et al. Mapping soil pH levels across Europe: An analysis of LUCAS topsoil data using random forest kriging (RFK)[J]. Soil Use and Management, 2023, 39(2): 900-916.

Q5. It is necessary to specify the method used to process the raw spectral data.

Thank you for your suggestions and feedback.In this paper, common preprocessing algorithms are operated on the dataset, such as SNV, MSC, etc. There is also no wavelength selection of the data. The main processing was done by screening the singular samples in the dataset using the MCCV algorithm. The screened singular samples are not used as training and testing samples for the model.

Q6. The title of Figure 2 also needs to be revised.

Thank you for your suggestions and feedback. This was an oversight on our part and has been changed to “Ghost Module Module.”

Q7. The conclusion section should present several issues that require further research.

Thank you for your suggestions and feedback. We have suggested questions for further research in the article based on the references[5]. Future research in this field will be directed towards the combination of spectral and remote sensing data for the integrated evaluation of intrinsic and extrinsic qualities of soil parameters.

[5] Hu S, Hong W, Xie J, et al. Non-destructive predictions of sugar contents in litchis based on near-infrared spectroscopy and stochastic configuration network[J]. Journal of Food Measurement and Characterization, 2024, 18(9): 8063-8077.

Additional Notes:

Due to the error of the code, there is an error in the calculation of the RMSE parameter in the previous experimental results. We have corrected the calculated errors, and the specific modified results are shown in red font in Table 3, Table 4 and Table 5 of the paper.

---

## [Decision Letter · Decision Letter 1]

Spectroscopic measurement of near-infrared soil pH parameters based on GhostNet-CBAM

PONE-D-24-54324R1

Dear Dr. Tian,

We’re pleased to inform you that your manuscript has been judged scientifically suitable for publication and will be formally accepted for publication once it meets all outstanding technical requirements.

Kind regards,

Himadri Majumder, Ph.D

Academic Editor

PLOS ONE

Additional Editor Comments (optional):

Reviewers' comments:

Reviewer's Responses to Questions

**Comments to the Author**

1. If the authors have adequately addressed your comments raised in a previous round of review and you feel that this manuscript is now acceptable for publication, you may indicate that here to bypass the “Comments to the Author” section, enter your conflict of interest statement in the “Confidential to Editor” section, and submit your "Accept" recommendation.

Reviewer #1: All comments have been addressed

Reviewer #2: All comments have been addressed

2. Is the manuscript technically sound, and do the data support the conclusions?

Reviewer #1: Partly

Reviewer #2: Yes

3. Has the statistical analysis been performed appropriately and rigorously? 

Reviewer #1: Yes

Reviewer #2: Yes

4. Have the authors made all data underlying the findings in their manuscript fully available?

Reviewer #1: Yes

Reviewer #2: Yes

5. Is the manuscript presented in an intelligible fashion and written in standard English?

Reviewer #1: Yes

Reviewer #2: Yes

6. Review Comments to the Author

Reviewer #1: The author has made reasonable revisions in response to the review comments and recommends accepting the paper

Reviewer #2: In this paper, we propose a

neural network algorithm using Ghostnet combined with Convolutional Block Attention

Module(CABM) to realize the near-infrared (NIR) PH spectral measurement of soil.

The issues have been addressed. I think it can be accepted.

7. PLOS authors have the option to publish the peer review history of their article (what does this mean?). If published, this will include your full peer review and any attached files.

Reviewer #1: No

Reviewer #2: **Yes: **Hongbiao Zhou

---

## [Editor Report · Acceptance letter]

PONE-D-24-54324R1

PLOS ONE

Dear Dr. Tian,

I'm pleased to inform you that your manuscript has been deemed suitable for publication in PLOS ONE. Congratulations! Your manuscript is now being handed over to our production team.

Kind regards,

on behalf of

Dr. Himadri Majumder

Academic Editor

PLOS ONE